# SARS-CoV-2 ORF8 as a Modulator of Cytokine Induction: Evidence and Search for Molecular Mechanisms

**DOI:** 10.3390/v16010161

**Published:** 2024-01-22

**Authors:** Marília Inês Móvio, Giovana Waner Carneiro de Almeida, Isabella das Graças Lopes Martines, Gilmara Barros de Lima, Sergio Daishi Sasaki, Alexandre Hiroaki Kihara, Emma Poole, Michael Nevels, Maria Cristina Carlan da Silva

**Affiliations:** 1Laboratório de Neurogenética, Universidade Federal do ABC (UFABC), São Bernardo do Campo, São Paulo 09606-070, Brazil; marilia.movio@ufabc.edu.br (M.I.M.);; 2Centro de Ciências Naturais e Humanas (CCNH), Universidade Federal do UFABC (UFABC), São Bernardo do Campo, São Paulo 09606-070, Brazil; giovana.carneiro@ufabc.edu.br (G.W.C.d.A.); lima.gilmara@ufabc.edu.br (G.B.d.L.); sergio.sasaki@ufabc.edu.br (S.D.S.); 3Division of Virology, Department of Pathology, Cambridge University, Level 5, Addenbrooke’s Hospital, Hills Road, Cambridge CB2 0QQ, UK; 4School of Biology, University of St Andrews, St Andrews KY16 9ST, UK; mmn3@st-andrews.ac.uk

**Keywords:** SARS-CoV-2, COVID-19, ORF8, cytokine storm

## Abstract

Severe cases of SARS-CoV-2 infection are characterized by an immune response that leads to the overproduction of pro-inflammatory cytokines, resulting in lung damage, cardiovascular symptoms, hematologic symptoms, acute kidney injury and multiple organ failure that can lead to death. This remarkable increase in cytokines and other inflammatory molecules is primarily caused by viral proteins, and particular interest has been given to ORF8, a unique accessory protein specific to SARS-CoV-2. Despite plenty of research, the precise mechanisms by which ORF8 induces proinflammatory cytokines are not clear. Our investigations demonstrated that ORF8 augments production of IL-6 induced by Poly(I:C) in human embryonic kidney (HEK)-293 and monocyte-derived dendritic cells (mono-DCs). We discuss our findings and the multifaceted roles of ORF8 as a modulator of cytokine response, focusing on type I interferon and IL-6, a key component of the immune response to SARS-CoV-2. In addition, we explore the hypothesis that ORF8 may act through pattern recognition receptors of dsRNA such as TLRs.

## 1. Introduction

Severe acute respiratory syndrome (SARS) coronavirus (CoV) type 2 (SARS-CoV-2), the causative agent of coronavirus infectious disease 2019 (COVID-19), is a member of the β-coronavirus family, along with SARS-CoV and Middle East respiratory syndrome (MERS) virus [1,2,3,4]. In the majority of healthy individuals, SARS-CoV-2 infection is asymptomatic or causes mild to moderate illness with symptoms such as fever, headache, cough, and breathlessness. However, severe cases develop acute respiratory distress syndrome (ARDS) and acute lung injury, leading to morbidity and mortality [2,3]. It is well-established that the severity of COVID-19 correlates with viral replication and hyper-responsiveness of the host immune system. The latter frequently involves an excessive production of cytokines by the host, referred to as a ‘cytokine storm’. This exacerbated cytokine response can lead to multi-organ failure and, in some cases, to a fatal outcome [5,6,7]. COVID-19 patients with severe disease, mainly those with specific co-morbidities and the elderly, have an exacerbated inflammatory response, evidenced by high levels of inflammatory markers (C-reactive protein, ferritin, D-dimer) in the blood, an increased neutrophil-to-lymphocyte ratio, high serum levels of pro-inflammatory chemokines and cytokines, such as interleukins IL-2, IL-6, IL-10, and IFN-γ, and low serum levels of type I and III interferons (IFNs) [8,9,10,11,12].

The SARS-CoV-2 genome is a positive-strand RNA that comprises 14 open reading frames (ORFs). ORF1a and ORF1b, the largest ORFs, are translated into polyproteins pp1a and pp1ab, which are cleaved to produce non-structural proteins Nsp1 to Nsp6. The other ORFs encode the structural proteins spike (S), membrane (M), envelope (E), and nucleocapsid (N), as well as the accessory proteins ORF3a, ORF4, ORF6, ORF7a, ORF7b, ORF8, ORF9b, and ORF10 [13]. The accessory proteins of coronaviruses are not considered to be primarily required for viral replication in vitro; however, they serve important functions during virus infection in vivo, contributing to immune evasion, cytokine induction, and enhanced virulence [14,15,16].

SARS-CoV-2 ORF8 is poorly conserved among other human coronaviruses but shares 95% amino acid sequence similarity with Bat-RaTG13-CoV, suggesting it originated from Bat-RaTG13-CoV ORF8 [17,18]. The ORF8 has high susceptibility to deletions and to point mutations which are associated with disease progression and outcome [19,20,21,22,23,24]. Notably, a mutation at position 84 that changes a leucine to a serine (ORF8 L84S) was the most frequent mutation in the first six months of the pandemic and underwent significant selection pressure [24,25]. This mutation has been related to mild disease outcome [23], similarly to SARS-CoV-2 strains that lack the ORF8 gene [26], indicating that ORF8 is a virulence factor. ORF8 is abundantly secreted both in vitro and in vivo, and it is highly immunogenic [27,28,29,30]. Among the accessory proteins of SARS-CoV-2, ORF8 has the largest protein interactome network [28], and several studies performed in the past few years have reported several biological properties of this protein, including immune evasion and signaling activation.

Here, we summarize research on SARS-CoV-2 ORF8 as a modulator of the cytokine response, in particular the pathways linked to type I IFN (IFN-α and IFN-β) and IL-6. ORF8 structure, evolution, roles in adaptive immunity, and overall functions have been extensively reviewed elsewhere [31,32,33,34,35,36]. We focus on ORF8’s capacity to modulate type I IFN and other cytokine responses, discussing the existing data. Additionally, we introduce new findings from our laboratory which indicate that the ORF8 L84S variant induces expression of IL-6 in presence of polyinosinic:polycytidylic acid (poly(I:C)) in human embryonic kidney (HEK)-293 and monocyte-derived dendritic cells (mono-DCs). Together with other, previous data, we hypothesize that ORF8 augments IL-6 production through Toll-like receptor signaling. Finally, we discuss the possible outcomes of cytokine induction for the host and the virus.

## 2. Cytokine Responses to SARS-CoV-2

Cytokine storm syndrome is characterized by a systemic inflammatory condition involving excessive circulating cytokine levels associated with endothelial damage, vascular permeability, coagulopathy, and infiltration of immune cells into tissues, which can consequently lead to multi-organ failure [37,38]. In severe cases of COVID-19, exacerbated cytokine dysregulation is a hallmark of the disease [39,40]. Clinical and experimental data indicate that there is an excessive production of pro-inflammatory IL-6, as well as TNF-alpha and IL-12, in SARS-CoV-2-infected patients, cells, and animal models [24,41,42,43].

Individuals with risk factors such as obesity, cardiovascular disease, acute kidney injury, pulmonary disease, male gender, impaired immunity, and old age are more likely to have an inappropriate immune responses and succumb to the disease [44,45]. Additionally, host genetics are implicated in disease outcome [46]. In particular, elderly individuals are more vulnerable due to decreased immunity and the age-related chronic pro-inflammatory status of their immune system, which increases tissue damage caused by the infection [47,48,49].

Importantly, SARS-CoV-2 is also associated with multisystem inflammatory syndrome (MIS) in pediatric patients, likely due to a post-viral immunological reaction to the virus [50]. Notably, there is increasing evidence pointing to a role of mitochondria in the pathogenesis of COVID-19. Damage of mitochondria occurs in age-related disorders, specially malfunctioning of the immune system, increasing pathogenesis in elderly individuals [51,52]. Moreover, alterations in mitochondria are associated with other disorders such as diabetes and obesity, impacting negatively the outcome of COVID-19 [53,54].

## 3. Immunity against SARS-CoV-2

Innate antiviral immunity is triggered by the recognition of viral pathogen-associated molecular patterns (PAMPs) or (DAMPs), released by damaged tissues, via cell pattern recognition receptors (PRRs), leading to the production of type I IFNs and pro-inflammatory cytokines. These cytokines function to inhibit viral replication and to regulate induction of adaptive immunity [55]. Type I IFNs constitute one of the first lines of defense against viruses. IFN production leads to induction of several IFN-stimulated genes (ISGs), via the janus kinase-signal transducer and activator of transcription (JAK-STAT) pathway, which leads to the induction of proteins which restrict viral replication in infected and neighbouring cells [56,57].

In RNA viruses, including SARS-CoV, SARS-CoV-2, and MERS virus, the double-strand RNA (dsRNA) generated during genome replication and transcription is sensed in the endosome by Toll-like receptors (TLRs) 3 and 7, or in the cytoplasm by the retinoic acid-inducible gene I (RIG-I)-like receptors (RLRs), RIG-I and/or melanoma differentiation-associated protein (MDA5) [58]. RIG-I and MDA5 interact with the adapter mitochondrial antiviral signaling (MAVS) protein, recruiting two IκB kinase (IKK)-related kinases, TANK-binding kinase 1 (TBK1) and inducible IκB kinase (IKKi). These kinases phosphorylate interferon regulatory factor (IRF) 3 and 7, leading to their dimerization and translocation to the nucleus, resulting in the activation of IFN-α/β expression. Additionally, MAVS recruits TANK1 via tumor necrosis factor receptor associated factor 6 (TRAF6) and activates the NF-κB signaling pathway, leading to cytokine production. Alternatively, sensing of viral RNAs by most TLRs, except for TLR3, involves the adaptor myeloid differentiation primary response gene 88 (MyD88). After TLR engagement, MyD88 forms a complex with IL-1 receptor-associated kinase (IRAK) family members, including IRAK1, IRAK2, and IRAK4, referred to as the Myddosome. IRAK1 associates with the RING-domain E3 ubiquitin ligase TRAF6, leading to activation of TAK1 followed by the activation of the NF-κB and mitogen-activated protein kinase (MAPK) pathways and the production of pro-inflammatory cytokines [59,60]. TLRs localized in endosomes (TLR3, TLR7, TLR8, and TLR9) activate NF-κB and IRF7. While TLR7, TLR8, and TLR9 use MyD88, TLR3 uses the adaptor TIR-domain-containing adaptor-inducing interferon-β (TRIF) and TLR4 uses both MyD88 and TRIF adaptors. TRIF binds to TRAF3, which then recruits the IKK-related kinases TBK1 and IKKε, activating IRF3 that mediates transcription of type I IFNs. Additionally, TRIF interacts with TRAF6 and promotes the activation of NF-κB and MAPKs [59,60].

Notably, recent studies have shown that RNA virus infection activates the cytoplasmic DNA sensor cGAS/STING by directly recognizing viral components or by sensing cellular DNA released from mitochondria or nuclei during cellular stress [61]. Accordingly, Neufeldt et al. (2022) showed that SARS-CoV-2 directs a cGAS-STING-mediated, NF-κB-driven inflammatory immune response in human epithelial cells that likely contributes to the inflammatory responses seen in patients [62].

The production of cytokines by the innate immune system, such as Type I IFN, TNF-alpha, IL-12, and IL-6, leads to the activation of adaptive immune defenses, resulting in the production of specific CD8+ cytotoxic T cells, CD4^+^ helper T cells, antigen specific B cells, and antibody production [63,64]. Currently, research indicates that CD4 “helper” T cells and CD8 “killer” T cells have a more protective role against the disease and that antibodies may play a secondary role in ultimately clearing SARS-CoV-2 [65]. The adaptive immune response works to control the viral infection and the level of damage caused by the cytokine storm [66], and subsequently the majority of individuals with SARS-CoV-2 remain asymptomatic or develop only mild symptoms [67,68,69]. However, in the case of failure to produce an adequate adaptive response to the persistent inflammation induced by innate immunity, the continuous cytokine storm can lead to multi organ infection and, potentially, organ failure [63,70]. One of the features of the uncontrolled immune response is an increase in innate immune cells and a high production of IL-6 associated with continued T cell activation that culminates in functional T cell exhaustion, loss of function, and, consequently, failure to eliminate the virus [71,72,73]. The importance of the adaptive immune response in controlling the cytokine storm is reflected in the fact that age-related immune senescence plays an important role in the development of the disease, mainly because of the reduction of de novo T cell responsiveness in elderly individuals [74,75,76].

In conclusion, SARS-CoV-2 infection can activate a strong innate immune system that leads to production of helper T cells (Th) and cytotoxic T cells (CTLs), Ref. [77], able to eliminate the virus. However, severe disease occurs when activation of the innate immune system fails to induce an adequate adaptive response to control the cytokine storm [63].

## 4. SARS-CoV-2 Antagonism of the Immune System

The viral antagonism of host immune responses is critical for virus replication and is an important player in the outcome of infection. Like other coronaviruses, SARS-CoV-2 has several mechanisms to usurp or inactivate both innate and adaptive host immune pathways [78,79,80,81]. During innate immune activation, the production of type I IFN is the first line of defense to limit viral replication and spread [82,83,84]. In response, several SARS-CoV-2 proteins employ strategies to efficiently block IFN induction and signaling pathways. As such, IFN responses in SARS-CoV-2-infected patients are weak and inadequate, reflecting the potent IFN antiviral antagonism [42]. Among the viral IFN antagonists are Nsp1, Nsp3, Nsp12, Nsp13, Nsp14, Nsp15, M, N, S, ORF3b, ORF6, and ORF8 [29,30,41,85,86].

The SARS-CoV-2 mechanisms to evade adaptive immunity are less understood; however, the virus appears to have strategies to combat T cell activation. One of them is the acquisition of mutations, which enable the virus to escape T cells’ responses. These mutations reduce the binding of viral antigens to HLA molecules and their subsequent presentation to T cells [87,88,89]. For instance, SARS-CoV-2 can readily alter its Spike protein via a single amino acid substitution which alters the Spike protein, preventing it from being recognized by CD8 T cells that target the most prevalent epitope in Spike [90]. Another important mechanism of viral adaptive immune evasion is down regulation of MHC I, which the virus achieves via the ORF8 protein [91].

## 5. SARS-CoV-2 ORF8 as an Immunomodulator

Among the SARS-CoV-2 accessory proteins, ORF8 is particularly remarkable as an immune system modulator. ORF8 interacts with the major histocompatibility complex class I (MHC-I), targeting it for degradation at lysosomes and causing SARS-CoV-2-infected cells to be more resistant to lysis by cytotoxic T-cells [91]. ORF8 has also been implicated in the escape of humoral immune responses. It binds to monocytes, causing a decrease in the levels of CD16 and a reduction in the ability to mediate antibody-dependent cellular cytotoxicity (ADCC) [92]. Additionally, its interaction with the human complement components C3/C3b and their metabolites leads to complement inhibitory activity [93]. In particular, the roles of ORF8 in modulation of type I IFN and other cytokine responses are discussed.

## 6. SARS-CoV-2 ORF8 as a Type I IFN Antagonist

In experiments performed in our laboratory, we observed that upon poly(I:C) stimulation, ORF8 L84S inhibits IFN-β in HEK-293 cells, while the same effect was not observed monocyte-derived dendritic cells (mono-DCs) (Figure 1C,E). In fact, we observed that ORF8 L84S can augment type I IFN-β expression via transfected poly(I:C) in mono-DCs (Figure 1E), as further discussed. Our findings corroborate several previous studies showing that ORF8 inhibit activation of IFN-β responsive promoter induced by Sendai virus [29] and poly IC in HEK-293 cells [94] and Hela cells [95]. This ability was found to be associated with a decrease in the nuclear translocation of IRF3 [94], and further evidence suggests that ORF8 causes deamidation of IRF3 via cellular CTP synthetase 1 (CTPS1), resulting in a loss of binding to IRF3-responsive promoters and reduced IFN expression [96]. Interestingly, in HeLa cells, ORF8 was shown to interact with heat shock protein 90 β family member 1 (HSP90B1), a molecular chaperone of ER, inhibiting its function and suggesting a further mechanism of IFN inhibition [95]. ORF8 can also affect IFN-β signaling, as demonstrated by its capacity to inhibit the IFN-stimulated response element (ISRE) upon treatment with IFN-β [29].

Furthermore, evidence indicates that ORF8 affects the IFN-β pathway in a cell-type-specific manner, since ORF8 84L inhibited expression of ISGs responsive to IFN-β via poly(I:C) stimulation, such as OAS3 and IFITM1 in HEK-293 but not A549 cells [97]. As such, our analyses indicate that while ORF8 84L inhibits IFN-β expression via poly(I:C) stimulation in HEK-293 cells, the same does not happen in monocyte-derived dendritic cells (mono-DCs) (Figure 1C,E).

Notably, in addition to antagonization of IFN-β production, ORF8 84L and L84S induce endoplasmic reticulum (ER) stress [94]. Indeed, induction of ER stress and activation of the Unfolded protein response (UPR) occurs during SARS-CoV-2 infection [98] and detailed analysis demonstrated that non-secreted ORF8 interacts with multiple ER chaperones such as BiP and calnexin, and with the ER stress sensors IRE1α, PERK, and ATF6 [91].

Furthermore, ORF8 expression in HEK 293T or HepG2 cells induces activation of UPR pathways. The ability to interact with ER proteins promotes ORF8 escaping from degradation, causing its accumulation in the ER lumen. Additionally, activation of the UPR regulates protein folding, remodels ER morphology, and accelerates protein trafficking. Importantly, recent studies show that there is direct crosstalk between the UPR and immune responses [99,100,101], and both studies described above observed concomitant inhibition of IFN-β with induction of ER stress and UPR in cells expressing ORF8 [94,102].

Significantly, in another study, SARS-CoV-2 infection was found to activate NF-kB protein expression and pathway activation in association with increased MAPK signaling and expression of the UPR inducer IRE-1α, suggesting a relationship between UPR signaling and NF-kB activity [101]. Consequently, additional studies are important to clarify the role of intracellular ORF8 in induction of UPR and inflammatory responses.

Finally, contrary to the previous findings indicating inhibition of IFN-β production and signaling, ORF8 did not inhibit activation of the IFN-β promoter by the RIG-I caspase recruitment domains (CARDs) [41,85] or the Sendai virus [41].

Overall, although several studies have pointed to a role of ORF8 in IFN-β antagonism, there are clearly inconsistences and the reasons for these discrepancies may be explained by differences in cell types, assay methods, assay timing after IFN-β stimulation, or amino acid variations in ORF8. Therefore, further experiments are necessary to clearly establish the possible role of ORF8 in the type I IFN antagonistic activity and to identify under which conditions it may occur. Arguably, if ORF8 has IFN regulatory activity, it may not be its primary function since there are also other SARS-CoV-2 proteins, such as ORF6, which is a well-established potent IFN-β antagonist [85].

**Figure 1 viruses-16-00161-f001:**
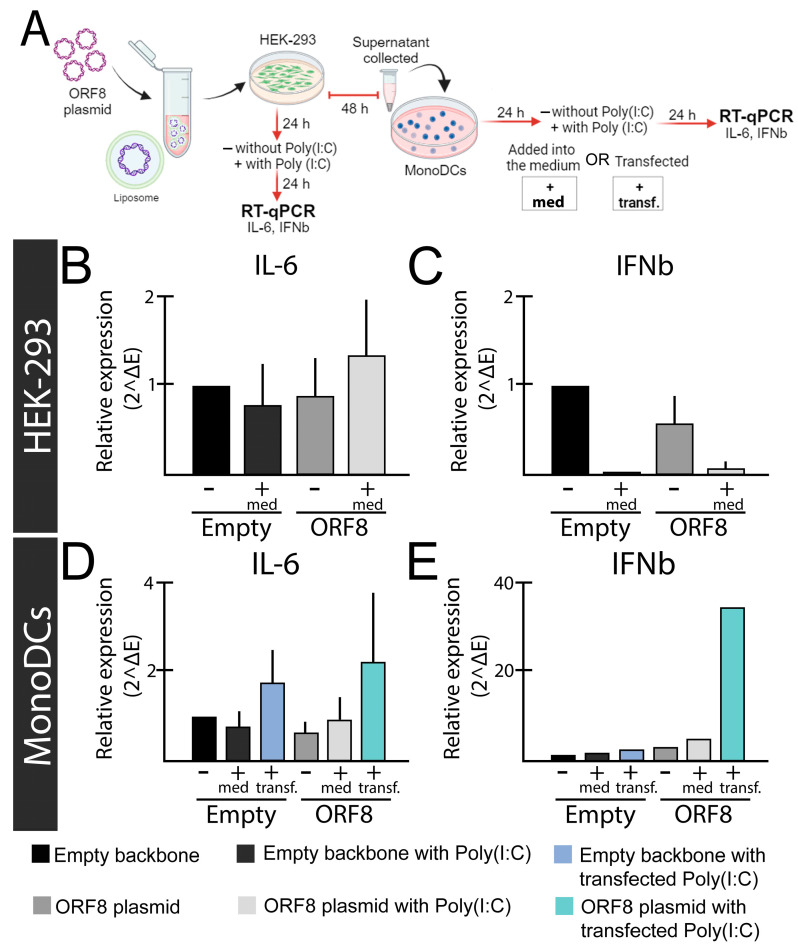
Relative expression levels of IL-6 and IFN-β genes after ORF8 expression. (**A**) Experimental design; (**B**,**C**) HEK-293 cells were transfected with ORF8 or empty plasmid (PCAGGS). To induce an immune response, Poly(I:C) (2 μg/mL) was added into the media (+med.) after 24 h of transfection with Lipofectamine™ 3000 (Invitrogen, Waltham, MA, USA). The relative expression of IL-6 (**B**) and IFN-β (**C**) were evaluated after 48 h of transfection. (**D**,**E**) Supernatants from HEK-293 cells transfected with ORF8 and empty vector were added onto mono-DCs isolated from blood according to Silva et al. (2207) [103], either alone (−), in combination with Poly(I:C) (2 μg/mL) in the media (+med.), or transfected (+transf.). After 48 h, cells were subjected to real time-quantitative PCR (rt-qPCR using SYBR green (Promega) and primers previously described [104]. Relative expression was determined using primers for the Glyceraldehyde 3-phosphate dehydrogenase gene (GAPDH) gene as previously described [105].

## 7. SARS-CoV-2 ORF8 as a Pro-Inflammatory Virokine

In addition to its role in immune evasion, plenty of evidence suggests that ORF8 acts as a virokine inducing pro-inflammatory responses. As mentioned, ORF8 is secreted from infected cells [30,106,107,108], and various studies indicate that the extracellular protein is able to activate inflammatory cytokines. This idea matches the observation that the presence of ORF8 in the plasma of infected individuals is conversely associated with survival [109,110].

Several studies have shown modulation of pathways linked to cytokine expression by ORF8 in vitro. Interactome and effectome analyses indicate that ORF8 interacts with the transforming growth factor (TGF)-β1-latent TGF-β binding protein 1 (LTBP1) complex, potentially dysregulating the TGF-β signaling pathway [111]. ORF8 was also shown to interact with the PRR nucleotide oligomerization domain (NOD)-like receptor family pyrin domain containing 3 (NLRP3) in CD14+/CD16+ monocytes, inducing a cytokine response. According to the proposed model, ORF8 enters monocytes through a non-receptor-mediated process and binds NLRP3 intracellularly, a process that needs to be further investigated [110].

Two subsequent studies indicate that ORF8 has the ability to mimic the pro-inflammatory human IL-17 cytokine and binds to RAW 264.7 in murine macrophages, as well as in human CD14+ THP-1 and U973 monocytes [30,112]. Similar to IL-17, ORF8 binds to the human hIL-17RA/C complex and induces heterodimerization and subsequent activation of downstream inflammatory responses. These responses include phosphorylation of p65 and IκΒα, as well as degradation of IκΒα and subsequent activation of the NF-κB signaling pathway (Figure 2). However, despite the similarities, signaling induced by ORF8 appears to have some differences from the IL-17 pathway because ORF8 induces expression of genes coding for pro-inflammatory factors of the IL-17 receptor pathway, including CCL20, CXCL1, CXCL2, and IL-6, in THP-1 and U937 cells but also causes overexpression of genes not known to be induced by IL-17, such as COL17A1, MMP10, and SERPINB2 [30]. Interesting, variations in ORF8 appear to affect its ability to induce cytokine expression, given that the ORF8 L84S variant, and, to a lesser extent, variants V62L and S24L, showed reduction in expression of CCL20, CXCL1, and IL-6 compared to wild-type ORF8 (84L), suggesting that these variants are associated with an attenuated inflammation phenotype [30].

In addition to the effects of mutations on ORF8 function, glycosylation appears to affect protein activity. Accordingly, X-ray crystallographic structure analysis of ORF8 indicates that four pairs of disulfide bonds and glycosylation at residue N78 are essential for stabilizing protein structure and that glycosylation regulates binding to monocytes [113], which appears to be required for cytokine induction [30]. It was demonstrated that secretion of glycosylated ORF8 occurs via the conventional method, through the Golgi apparatus, while unglycosylated ORF8 is secreted via unconventional pathways. However, only the unglycosylated ORF8, due to a mutation at the N78 residue, was able to bind to the IL-17RA receptor and induce cytokine expression in macrophages or mice in one study [107]. Conversely, in another study, glycosylated ORF8 (purified from HEK-293 culture supernatants) induced inflammation, whereas the unglycosylated form (produced in *E. coli*) did not. However, in the same work, ORF8 induced high levels of cytokine synthesis, even when protein glycosylation levels were reduced with Brefeldin A or Monensin treatment [30]. Therefore, further studies are required to clarify the role of glycosylated versus unglycosylated forms of ORF8 regarding differential functions and relevance during infection, especially since glycosylated ORF8 seems to be predominantly secreted in supernatants of SARS-CoV-2-infected cells and sera of COVID-19 patients [110].

Importantly, as described above, non-secreted ORF8 was shown to interact with ER stress sensors and induce activation of UPR pathways and possibly inflammatory response [94]; therefore, both secreted and non-secreted forms may be able to induce cytokine expression in different ways.

## 8. SARS-CoV-2 ORF8 as a Potential Cytokine Modulator through TLRs

Intriguingly, a recent study by Ponde et al. (2023) challenges previous findings in myeloid cells [30,112], showing that IL-17RA and IL-17RC are not required for ORF8 signaling in macrophages and monocytes but that it depends on the TLR/IL-1 family adaptor MyD88, indicating that it occurs though TLR recognition (Figure 2) [114]. The authors argue that the co-immunoprecipitation and ligation assays performed in previous studies do not prove a direct interaction between the IL-17 receptor and ORF8 [30,112]. Therefore, additional experiments may be required to convincingly demonstrate binding to and signaling through the IL-17 receptor by ORF8.

Our own observations demonstrated that secreted ORF8 L84S expressed in HEK-293 cells augments expression of IL-6 in HEK-293 cells and mono-DCs as measured with RT-qPCR (Figure 1B,D) and enzyme-linked immunosorbent assay (ELISA) (Figure 3). As mentioned earlier, we also observed induction of IFN-β in mono-DCs (Figure 1E), but not in HEK-293 cells (Figure 1C) in the presence of poly(I:C). Notably, ORF8 alone (without poly(I:C) stimulation) did not induce cytokine expression in any cells (Figure 1).

In agreement with our findings, as reported in by Kriplani et al. [108], ORF8 L84S variant, but not ORF8 L84, induces IL-6 production in primary monocyte-derived macrophages (MDMs) in the presence of poly(I:C). However, neither ORF8 84L and L84S combined with poly(I:C) induced or inhibited IFN-β production. This observation appears to be at odds to our findings and may be explained by differences in cell types. Although we did not test ORF8 L84 in our assays, these findings suggest that ORF8 L84S can induce IL-6 production in mono-DCs and MDMs, likely in a sequence-specific manner. We believe that the most plausible explanation for these findings is that ORF8 acts through one or more cellular PRRs which sense dsRNA (poly(I:C)), such as TLR3, RIG-I, or MDA5 [115], increasing IFN-β and IL6. TLR3 localizes to both the plasma membrane and the endosome in HEK-293 cells [116] 64, 65, and 71 but is exclusively intracellular in immature dendritic cells (iDCs) [117]. Coupled with our observation that activation of IL-6 occurs in HEK-293 cells with media-supplemented poly(I:C), but only upon transfection in mono-DCs (Figure 1B,D); we hypothesize that ORF8 is acting through TLR3 (Figure 2). However, the involvement of other PPRs that sense the SARS-CoV-2 RNA genome cannot be ruled out.

Consistent with this idea, the work by Ponde et al. (2023) demonstrated that ORF8 signaling depends on MyD88, the canonical adaptor for inflammatory signaling downstream of several TLR family members (TLR2, TLR4, TLR5, TLR7, TLR8, and TLR9), except TLR3, which uses the adaptor TRIF [114] (Figure 2). However, recent evidence indicates that TLR3 might nonetheless use, at least in certain circumstances, the MyD88-mediated pathway [118].

Interestingly, it is not uncommon for viral proteins to enhance poly(I:C)- and viral dsRNA-induced TLR3 signaling, as exemplified by polymerases from two genotypes of hepatitis C virus and three viral capsid proteins (1bD21, 2a.m26–30, B-cp, Rcp, and H-cp183). These proteins have the common ability to bind dsRNA [119]. It is possible that viral proteins, released by cell lysis or secreted from infected cells, act as PAMPs binding to RNA and/or cell surface or intracellular receptors after being taken up by cells, subsequently activating innate immune responses.

Based on the available evidence, including our new findings, we believe that it is necessary to investigate whether TLR3s or other PRRs are required for ORF8-activated innate immune signaling.

## 9. Conclusions, Challenges, and Future Prospects

As discussed, there is plenty of evidence that ORF8 is a modulator of cytokine responses during SARS-CoV-2 infection. ORF8 has the capability to induce inflammatory responses, yet conversely, several studies indicate that it inhibits type I Interferon, even though not all studies have consistently supported this inhibition effect. These two activities diverge from one to another between studies and the precise mechanisms, by which ORF8 functions are still not known. It is possible that ORF8 acts as an anti- and pro-inflammatory factor in different phases of infection, at early and late stages, respectively, and more studies need to be done to unveil the mechanisms of these observations. One important question is if the release of secreted ORF8 activating inflammation has any benefit for the virus or if it is just a consequence of viral replication contributing to immune pathology associated with viral infection. 

During viral infection, many viral glycoproteins, viral RNA, and other viral proteins are recognized by cellular PRRs, stimulating cytokine and chemokine production, and consequently, the virus has several strategies to block this response. The outcome of the interplay between host immunity and viral countermeasures dictates the degree of virulence as well as the nature of the immune response. The antiviral effects of cytokines are important in controlling viral infection and for host survival. Moreover, viral induction of cytokines inhibiting viral replication may also represent a strategy that viruses have evolved to reduce their visibility to the immune system and, therefore, promote viral persistence by escaping immune recognition. There is plenty of evidence to suggest that ORF8 has the capability to induce inflammatory responses, mainly IL-6, likely contributing to ‘cytokine storm’-related disease. In this regard, it will be important to elucidate if ORF8 indeed acts as a virokine to modulate immunity upon its secretion, as in the case of activation of the IL-17 RA cascade [30,112]. It is possible that ORF8 also acts as a secreted PAMP that is recognized by cell PRRs, activating inflammatory responses independently of cellular infection. In fact, other SARS-CoV-2 proteins might function as PAMPs and induce signaling through TLRs, such as the S1 subunit [120] and the E protein [121], which induce neuroinflammatory responses through TLR4 and TLR2, respectively.

Notably, in our studies, ORF8 only induced IL-6 production in the presence of poly IC. One might speculate that when high levels of viral replication are achieved, secreted ORF8 (ORF8 L84S) is able to induce cytokine expression, in the presence of viral RNA released by infected cells, in order to control the virus and benefit viral persistence by escaping immune recognition, a hypothesis that requires more investigation.

We have commented here about the current data indicating that ORF8 interferes with the immune response, and we pointed to a number of unanswered questions that might be addressed in future studies. To summarize, one of the major challenges in virology is to clearly understand virus-induced signaling and how this affects viral replication and the host-induced disease. In the case of SARS-CoV-2 ORF8, much work has been conducted, showing several biological properties of this intriguing protein. However, future studies are necessary to define its precise role in the pathogenesis of COVID-19.

## Figures and Tables

**Figure 2 viruses-16-00161-f002:**
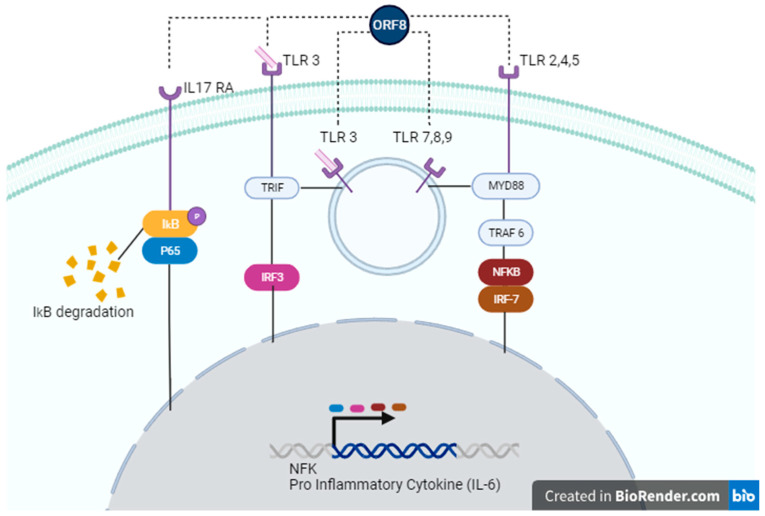
Possible signaling pathways activated by ORF8. ORF8 accessory protein can act through IL-17 RA (**left**), TLR 3 (**left**, **center**), TLR 7,8,9 (**center**), or TLR 2,4,5 (**right**). In the IL-17 RA cascade, ORF8 leads to IkB/P65 phosphorylation, causing degradation of IκΒα and subsequent activation of the NF-κB signaling pathway. Recognition of ORF8 by TLRs activates TRIF and/or MyD88 pathways, enhancing gene transcription of NF-κB genes or pro-inflammatory cytokines, such as IL-6.

**Figure 3 viruses-16-00161-f003:**
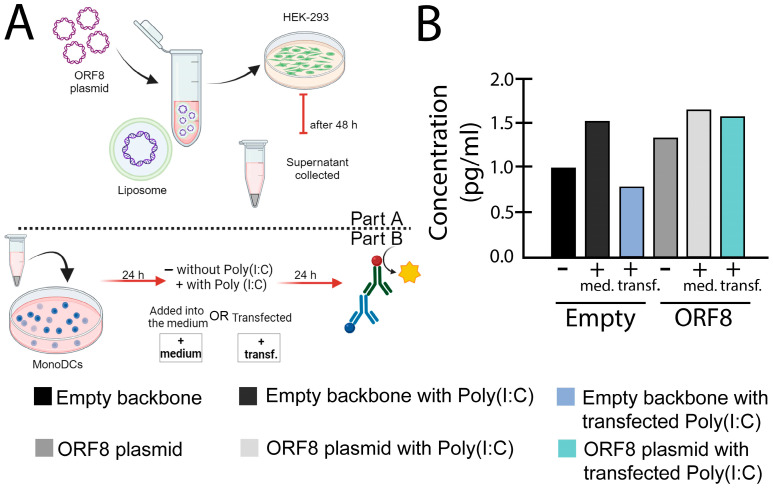
IL-6 levels induced by ORF8 measured with enzyme-linked immunosorbent assay (ELISA) assay. (**A**) Experimental design used for the experiment. (**B**) After 24 h post transfection, supernatants from HEK-293 cells transfected with ORF8 and empty vector (PCAGGS), using Lipofectamine™ 3000 (Invitrogen, Waltham, MA, USA), were added onto mono-DCs, either alone (−), in combination with Poly(I:C) (2 μg/mL) in the media (+med.), or transfected (+transf.). After 24 h, the supernatants were collected and subjected to ELISA (R&D Systems) for IL-6. The bars represent the mean value normalized by the control (empty supernatant).

## Data Availability

Not applicable.

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
