# Peer review of "SARS-CoV-2 ORF8 as a Modulator of Cytokine Induction: Evidence and Search for Molecular Mechanisms"

_viruses, 2024, doi:10.3390/v16010161_

Round 1

Reviewer 1 Report

Comments and Suggestions for Authors

Not sure what the author’s are trying to do with this paper, other than get a bit of their own unpublished data published – so it is really a review? Should the authors get more data and then submit a data-based article? There are better and more comprehensive reviews already published on ORF8 (indeed, the authors reference them). In particular, it lacks a unifying underlying hypothesis/bit of original insight/theme that would make it more valuable. The bulk of the paper, in terms of grammar and general writing, is actually quite good – although the abstract has several basic grammatical and English errors in it. Would suggest at least improving both the abstract, introduction and concluding remarks in light of the above. In short, the authors need to position it better,  plus:

1)      The authors need to tackle why the virus would modulate the immune system in this way (why does it apparently enhance the inflammatory response – this is not simply a “reaction” of the host, but a programmed mechanism that must benefit the virus in some way). Does their little bit of new data help in understanding this? What is the evolutionary role here? This would help the reader to position what the authors are thinking. Provide some insight. There are some big clues here which the authors could tease out. One of them is how it controls the UPR, for instance.  

2)      There is a big omission about the specific populations who seem to succumb to the virus, and this could well be a key clue, which could potentially make the paper much more interesting and informative: older, and people with co-morbidities, especially the metabolic syndrome (obesity), appear to be more susceptible. One of the key underlying findings in these populations is an element of chronic inflammation, which at least in part, is related to a reduction in mitochondrial function – they already exhibit an immune imbalance. This obviously ties into immunosenescence. Viruses have to modulate metabolism as well as control the immune system.

3)      In contrast, what about children who seem to develop an inflammatory spiral?

Comments on the Quality of English Language

See above. 

Author Response

1-Not sure what the author’s are trying to do with this paper, other than get a bit of their own unpublished data published – so it is really a review? Should the authors get more data and then submit a data-based article? There are better and more comprehensive reviews already published on ORF8 (indeed, the authors reference them).

Answer: We agree that there are extensive reviews about SARS-CoV2 ORF8, as the reviewer correctly observes, we have cited them. However, the aim of this review is to focus on the ability of ORF8 to function, particularly, as an immune system modulator. We gathered all the available published data on this aspect of ORF8 function and the main point of the review is to show that, despite all the work that has been done, the precise mechanisms of cytokine induction by ORF8 are not clear. In addition, we believe that our piece of data provides important information which adds to our knowledge of immune functions of ORF8 and opens future research to clarify these questions.

Regarding the data included in the review, before submission we contacted the assistant Editor Mina Liu and she suggested we submit in this format.

2- In particular, it lacks a unifying underlying hypothesis/bit of original insight/theme that would make it more valuable.

Answer: Our hypothesis is that: that ORF8 likely acts through different pathways to induce cytokine responses. In particular, based in our work and work of the others we believe that ORF8 acts through one or more TLRs and therefore more research is required to fully elucidate the function of ORF8 cytokine induction pathways.  We have added that to the text to clarify this point:

- Based on the available evidence including our new findings, we believe that it is necessary to investigate whether TLR3s or other PRRs are required for ORF8-activated innate immune signaling.

  • It is possible that ORF8 also acts as secreted PAMP, that is recognized by cell PRRs, activating inflammatory responses independent of cellular infection and that needs to be investigated

3- the abstract has several basic grammatical and English errors in it. Would suggest at least improving both the abstract, introduction and concluding remarks in light of the above. In short, the authors need to position it better,  

Answer: We corrected the english errors and tried to improve the the abstract, introduction and concluding remarks

4- The authors need to tackle why the virus would modulate the immune system in this way (why does it apparently enhance the inflammatory response – this is not simply a “reaction” of the host, but a programmed mechanism that must benefit the virus in some way). Does their little bit of new data help in understanding this? What is the evolutionary role here? This would help the reader to position what the authors are thinking. Provide some insight. There are some big clues here which the authors could tease out. One of them is how it controls the UPR, for instance.  

Question: why the virus would modulate the immune system in this way (why does it apparently enhance the inflammatory response – this is not simply a “reaction” of the host, but a programmed mechanism that must benefit the virus in some way). Answer: We discuss this important point in the section: Conclusions, Challenges, and Future Prospects.

Regarding the activation of the UPR, it is published that non-secreted ORF8 activates the ER stress response and UPR. We added comment that in the section: SARS-CoV-2 ORF8 as a pro-inflammatory virokine (bellow)

“In addition, as described above non-secreted ORF8 was shown to interact with ER stress sensors and induce activation of UPR pathways and possibly inflammatory response suggesting therefore that both secreted and non-secreted forms are able to induce cytokine expression in different ways.

5- There is a big omission about the specific populations who seem to succumb to the virus, and this could well be a key clue, which could potentially make the paper much more interesting and informative: older, and people with co-morbidities, especially the metabolic syndrome (obesity), appear to be more susceptible. One of the key underlying findings in these populations is an element of chronic inflammation, which at least in part, is related to a reduction in mitochondrial function – they already exhibit an immune imbalance. This obviously ties into immunosenescence. Viruses have to modulate metabolism as well as control the immune system.  In contrast, what about children who seem to develop an inflammatory spiral?

Answer: We have complemented the section: Cytokine responses to SARS-CoV-2 with this important data.  

Reviewer 2 Report

Comments and Suggestions for Authors

One of the important problems for virology is understanding how the virus is able to overcome the mechanisms of innate immunity. This problem is relevant in the case of the SARS-CoV-2. The article by Marilia Inês Móvio and co-authors raises exactly this question, namely the participation of the ORF8 protein in the mechanisms of suppression of innate immunity.

The work carried out within the framework of the article undoubtedly deserves attention. The authors made a detailed review of the already available information regarding this viral protein. It is very good that the review allows you to get information quite clearly. There is no unnecessary information.

I believe that the article can be published in its current form.

Author Response

The work carried out within the framework of the article undoubtedly deserves attention. The authors made a detailed review of the already available information regarding this viral protein. It is very good that the review allows you to get information quite clearly. There is no unnecessary information.  I believe that the article can be published in its current form.

Answer:We thank the reviewer and modified the manuscript according to the others reviewer's comments 

Reviewer 3 Report

Comments and Suggestions for Authors

Comments on the Quality of English Language

Author Response

1.Abstract: 

The abstract can be improved.  

- Can you ameliorate the expression “resulting in damage to the lungs and other organs”, that is generic and scientifically imprecise.

Answer: We abstract has been modified in order to improve it.

-Please explain better your study design in the abstract

Answer: We tried to provide enough detail about our data in the abstract.  

2.Introduction:

- The introduction is appropriate.

Answer: Thank you

-Please add a sentence to explain the role of ORF8 in escaping/damaging the adaptive immunity

Answer: Wee added data about the adaptive immunity in the section: Immunity against SARS-CoV.

We also added data about how the virus subverts the adaptive immunity in the section: SARS CoV2 antagonism of the immune system

3.Methods:

-Methods are missing: how was the mini-review conducted? Is it a scoping review? Please provide a detailed explanation. You can choose to add a paragraph either in the introduction or separately.

Answer: We have described the methods in more detail in the figure legends.

Regarding the data included in the review, before submission we contacted the assistant Editor  and she suggested we submit in this format.

4.Cytokine responses to SARS-CoV-2:

- Can you show, by a picture, the mechanisms by which SARS-COV MERS viruses activate IFN and cytokines productions?

Answer:  While we agree that it could be informative to compare IFN and cytokine production by SARS-COV MERS viruses we feel this is beyond the remit of the current manuscript.

5.

-Line 136 “more resistant to lysis non less

Answer: Corrected

- Lines 135-142→ I suggest you to add a separate paragraph titled “others ORF8 strategies to evade the immune system”, or something similar, because the content does not match the title, in this part. Moreover, I would mention it after in the test.

Answer: We have modified the titles 

6.SARS-CoV-2 ORF8 as a type I IFN antagonist:

Here I suggest to show your study results here and starting to comment them after.

- This paragraph is difficult to read because it is written in the form of a list without an organized and engaging exposition of the subject. Initially, I suggest discussing your findings in relation to the data present in the literature. Subsequently, present in a more integrated and discursive manner what you have mentioned in the paragraph.

Answer: We have started to show our results and modified the section in order to discuss our findings in relation to the current data.

7.SARS-CoV-2 ORF8 as a pro-inflammatory virokine:

- The content is interesting. I recommend making it easier to read by using shorter sentences and connecting them with a clear mental scheme.

Answer: We modified the section in order to make easier to read. 

8.SARS-CoV-2 ORF8 as a potential cytokine modulator through TLRs

- As previously mentioned, strive to improve readability by incorporating shorter sentences and creating a coherent mental flow.

Answer: We modified the section in order to improve it.

9.Concluding remarks:

- Please summarise this paragraph and add some future perspectives about the issue.

Answer: We rewrote concluding remarks in order to improve it

Reviewer 4 Report

Comments and Suggestions for Authors

Check the italics of in vitro and in vivo

I advise the authors to generate two articles: an original article explaining the methodology in detail and given the appropriate scientific rigor and another article (review) after the publication of their original paper.

Throughout the text, they should put more figures.

Comments on the Quality of English Language

Minor editing of English language required

Author Response

I advise the authors to generate two articles: an original article explaining the methodology in detail and given the appropriate scientific rigor and another article (review) after the publication of their original paper.

Answer: We thank the reviewer for their suggestions. The paper has been submitted in the current format as recommended by the Editor at Viruses, which we agree is suitable for this manuscript.

The methodology was described in more detail in the figure legends.

Round 2

Reviewer 3 Report

Comments and Suggestions for Authors

Thank you for the review. Most of the highlighted points have been consistently implemented. I have no further comments.

Reviewer 4 Report

Comments and Suggestions for Authors

The authors covered all aspects.